# Modeling the Dynamics of Protein–Protein Interfaces, How and Why?

**DOI:** 10.3390/molecules27061841

**Published:** 2022-03-11

**Authors:** Ezgi Karaca, Chantal Prévost, Sophie Sacquin-Mora

**Affiliations:** 1Izmir Biomedicine and Genome Center, Izmir 35340, Turkey; ezgi.karaca@ibg.edu.tr; 2Izmir International Biomedicine and Genome Institute, Dokuz Eylul University, Izmir 35340, Turkey; 3CNRS, Laboratoire de Biochimie Théorique, UPR9080, Université de Paris, 13 rue Pierre et Marie Curie, 75005 Paris, France; chantal@ibpc.fr; 4Institut de Biologie Physico-Chimique, Fondation Edmond de Rothschild, PSL Research University, 75006 Paris, France

**Keywords:** protein interactions, protein interfaces, protein dynamics, molecular modeling, protein docking

## Abstract

Protein–protein assemblies act as a key component in numerous cellular processes. Their accurate modeling at the atomic level remains a challenge for structural biology. To address this challenge, several docking and a handful of deep learning methodologies focus on modeling protein–protein interfaces. Although the outcome of these methods has been assessed using static reference structures, more and more data point to the fact that the interaction stability and specificity is encoded in the dynamics of these interfaces. Therefore, this dynamics information must be taken into account when modeling and assessing protein interactions at the atomistic scale. Expanding on this, our review initially focuses on the recent computational strategies aiming at investigating protein–protein interfaces in a dynamic fashion using enhanced sampling, multi-scale modeling, and experimental data integration. Then, we discuss how interface dynamics report on the function of protein assemblies in globular complexes, in fuzzy complexes containing intrinsically disordered proteins, as well as in active complexes, where chemical reactions take place across the protein–protein interface.

## 1. Introduction

Protein–protein interactions (PPIs) lie at the heart of the machinery of life. Numerous cell mechanisms, such as metabolic pathways, transport or immune response, rely heavily on protein interaction networks [1,2,3]. As a consequence, dysfunctional PPIs are often linked to diseases [4], putting PPIs out as a central target in drug discovery. Protein–protein assemblies cover a wide range of size and shapes [5,6,7,8], from simple dimers to the large viral capsids that are formed by over a thousand protein chains [9]. Protein complexes can also be distinguished based on their lifetime in the cell, from weak transient complexes with micromolar binding affinities that will only last a few seconds, to permanent complexes with nanomolar binding affinities [10].

Experimentally, a wealth of data regarding the protein interactome can be acquired from proteomics [11,12]. Atomistic resolution information on macromolecular assemblies, on the other hand, is gathered by X-ray crystallography, NMR spectroscopy, and cryo-electron microscopy (cryo-EM) techniques (as deposited in the Protein Data Bank (PDB)) [13]. Recent advances in cryo-EM techniques in particular have put the structural determination of large molecular machines within our reach [14,15]. The interested reader will find a detailed listing of the various biophysical methods used for the detection of PPIs, listing their advantages and disadvantages, in a recent review by Zhou et al. [16].

In parallel with experimental techniques, in silico approaches have also evolved to provide a promising complementary strategy in both predicting the interaction partners [17,18,19] and determining the three-dimensional (3D) structure of protein complexes [20,21,22]. In recent decades, the most prominent tool to model protein complexes has been *docking*, where one attempts to determine the structure of a protein complex starting from its individual partners. Established in 2001, the CAPRI (Critical Assessment of predicted Interactions) initiative has fostered significant developments in docking and scoring methods [23,24,25]. In recent years, CASP (Critical Assessment of Predicted Interactions) has also introduced an *assembly* category in order to combine both worlds, i.e., tertiary and quaternary structure prediction methods [26,27,28]. In both CAPRI and CASP, the standard assessment criteria used to validate protein complex models rely on a single reference X-ray structure of the target complex. This view endorses the image of a static protein–protein interface, which has been increasingly questioned over recent years. We now have a considerable amount of data indicating that protein interfaces are dynamic, presenting conformational heterogeneity, especially when they include disordered, flexible segments [29,30,31,32,33,34]. Within the limits of the current docking strategies, one could deal with conformational flexibility upon starting from an ensemble of structures or performing a short interface refinement. However, none of these aspects provide enough sampling to report on the stability and specificity of the complexes. Therefore, new approaches should be developed to take functional interfacial dynamics into account to model stable and specific protein interactions [35,36,37,38,39,40].

To aid the development of such tools, in this review, we first focus on the recent strategies established to investigate protein–protein interfaces in a dynamic perspective (which are summarized in Figure 1), including enhanced sampling, multi-scale modeling, and experimental data integration. The second half of the manuscript discusses recent work where interface dynamics report on function in the case of globular protein complexes, fuzzy complexes encompassing intrinsically disordered partners, and active complexes, where chemical reactions take place across the protein–protein interface.

## 2. Tools for Calculating PPI Dynamics

### 2.1. Classical All-Atom Molecular Dynamics Simulations

Classical molecular dynamics (MD) simulations remain a first-choice tool in understanding the dynamics of biomolecular assemblies [41,42]. MD-produced trajectories do not only provide us with dynamic structural data on protein interfaces, but they also enable us to characterize the conformational space of the whole complex. Information on interface dynamics can be used to locate transient functional features, such as pockets, that would elude X-ray crystallography. Such sites can be exploited as binding sites for drug screening to modulate protein–protein interactions [43,44]. So, within this context, MD simulations can help us gain a deeper understanding of the macromolecular mechanisms taking place in the cell. Thanks to recent advances in hardware technologies, especially in graphics cards, the last ten years have seen impressive achievements in the extent of complexes simulated by MD [45]. One such example is the all-atom simulation of a complete ribosomal structure during the translocation process [46]. As a result, the *computational microscope* [47] offered by MD simulations has become a standard weapon to find where to target pathogen proteins, with recent examples including the simulation of protein assemblies from the Ebola virus [48] and diverse simulation studies on the interaction between the SARS-CoV-2 spike protein and the human ACE2 receptor [49].

### 2.2. All-Atom Enhanced Interface Sampling Approaches

The efficient sampling of the rugged conformational landscape of protein interfaces is an expensive process. Despite advances in hardware technologies, the computational cost of simulating long time scales (multiple microseconds) to escape local energy minima can still present a limiting factor for many systems. These considerations led to the development of enhanced sampling strategies, focusing only on the interface sampling over time [50]. For example, Peiffenberger and Bates [51] used metadynamics [52,53] simulations within the contact map space (CMS). The CMS is built from the inter-residue contacts observed between the receptor and ligand proteins deduced from the initial docked structure. This procedure is used to refine protein–protein complex interfaces. The discrete molecular dynamics (DMD) approach, where particles are assumed to move at a constant velocity from collision to collision, was used by Emperador et al. [54] to relax protein–protein interfaces. This approach greatly reduces the calculation costs, while improving the poses provided by rigid-body docking. Siebenmorgen et al. [55,56] introduced a repulsive bias, keeping the ligand protein at different distances from the receptor in a classic replica exchange scheme (RS-REMD). This strategy accelerates the searching process for the correct binding site and enabled them to identify the native binding site and calculate binding affinities for protein complexes. Scafuri et al. [57] used scaled molecular dynamics (SMD) to rank protein–protein docking poses that were initially produced by the rigid-body docking procedure of HADDOCK [58]. In SMD, the potential energy is multiplied by a scalar, λ < 1 [59], where the resulting weakening of the forces between the proteins induces perturbations, permitting us to identify the most stable docking poses.

### 2.3. Sampling the Protein Interface at the Coarse-Grain Level

In docking, most approaches treat the interacting protein partners as rigid bodies during the first exploratory steps. The backbone flexibility is only addressed at later refinement stages. In these approaches, whereas side-chain flexibility can be addressed to a certain extent, the backbone flexibility remains a costly challenge [60]. To reduce this cost and enable the conformational sampling of backbone flexibility, coarse grain models are used [61]. In this perspective, Kurcinski et al. [62] recently combined the CABS CG protein model with Replica Exchange Monte Carlo (REMC) simulations, thus allowing the sampling of large structural backbone conformational changes across a protein–protein complex. The PACSAB coarse-grained force field [63] of Emperador et al. was specifically developed to include conformational variations in many protein systems. Combined with an implicit solvent model and DMD simulations [64], it was used to filter docking poses of protein complexes, as nonnative poses tend to deviate significantly from their initial conformation through time, thus leading to a complete disruption of the ligand-receptor complex. Membrane protein complexes are particularly expensive to model at the atomic scale, as they require the explicit modeling of their lipid environment. Therefore, coarse graining has been used quite often in membrane protein complex modeling [65]. For example, Liao et al. [66] used a combination of all-atom, hybrid, and coarse-grain (with the MARTINI model [67]) representations with MD simulations to study the dynamics of a complex formed by two G protein-coupled receptors embedded in a lipid bilayer.

Another option when trying to address the interface flexibility with a reduced computational cost is to use elastic network models (ENM) [68,69,70]. ENMs can help identify the collective low-energy modes controlling the structural fluctuations around a reference conformation. Zen et al. [71] used this approach to characterize the dynamics of the interface in a set of 22 protein dimers (comprising both obligate and non-obligate complexes). They showed that the mobility of the amino acids located at the dimeric interface is generally lower than for the amino acids on the rest of the surface. Stadler et al. [72] calculated the mechanical properties of the hemoglobin tetramer using a coarse-grain ENM and showed how the amino acids presenting the largest mechanical variability when comparing the human and chicken tetrameric Hb structures are located at the central interface of the assembly.

ENMs are also commonly used to perform normal modes analysis (NMA) and investigate the collective dynamics in a protein structure. For example, Liang et al. [73] studied various states (active and autoinhibitory) of DNMT3A in dimeric and tetrameric assemblies to determine its intrinsic dynamics and showed how the central interface infers allosteric properties. Tsuchiya et al. [74] used a protein representation based on the dihedral angle space and NMA [75] to study the interface dynamics in over 500 homodimers.

### 2.4. Integrating Experimental Data Reporting on the Protein Interface Dynamics

Next to the portrayed computational tools, several biophysical techniques can provide information on the conformational heterogeneity of the interaction partners. The structure and dynamics of interface residues, or the lifetime of local contacts, can be obtained via X-ray crystallography, NMR, cryoEM, or FRET [38]. NMR and all-atom MD is a classic combination to study protein assemblies, with NMR parameters being used to set up the starting structures for the simulations [76]. Although the use of NMR is limited by the molecular weight of the complex (as it should be <30 kDa), recent developments have made the characterization of dynamic complexes far more accessible [6,77]. Solvent paramagnetic relaxation enhancement (sPRE) experiments, which use NMR with the addition of soluble paramagnetic molecules, will provide quantitative information regarding surface accessibility at atomic resolution. These data can be used to map solvent-exposed regions in protein assemblies and allows the detection of transient interactions in fuzzy complexes [78]. Similar data can be obtained from limited proteolysis [79] and H/D exchange [80] mass spectrometry experiments. When coupled with protein cross linking, mass spectrometry can also help to localize flexible protein regions that cannot be resolved with cryo-EM or X-ray [81]. SAXS [82] or mass spectrometry [83], on the other hand, can provide data about the stoichiometry and shape of the assembly. A recent work by do Nascimento et al. combined a quartz crystal microbalance and dual polarization interferometry for the real-time investigation of the thermodynamics and conformational dynamics of the TRIM12 antibody–antigen complex [84]. All this information can be converted in spatial restraints that are used for sampling in integrative modeling platforms [85,86,87], in combination with modeling tools (docking, NMA, Monte Carlo, or MD simulations).

For example, Kharche et al. investigated the interaction between the CXC chemokine receptor 1 (CXCR1) and its cognate chemokin, interleukine-8 (CXCL8) [88]. Here, Kharche et al. used MD simulations with the coarse-grain MARTINI force-field, which enabled us to run microsecond-long simulations for this large system, comprising a receptor protein embedded in a lipid bilayer. These were complemented by shorter all-atom MD simulations, as well as by NMR chemical shifts. These were obtained for the receptor N-terminal region, which is located across the protein–protein interface, and compared to values from NMR experiments. This global strategy permits us to investigate the impact of ligand binding on the receptor interface dynamics.

Interestingly, coevolutionary data retrieved from the sequence of interacting proteins can also bring precious information when investigating the dynamics of protein–protein interfaces, as they can be used to guide the assembly formation [89]. Direct coupling analysis [90] of over 13,000 sequences was used in combination with site-directed mutagenesis and MD simulation by Dago et al. to predict interdomain contacts and multiple conformations in the histidine kinase autophosphorylation complex [91,92]. Malinverni et al. developed an approach combining molecular simulations based on both coarse-grained and atomistic models with coevolutionary sequence analysis to shed light on a transient HSP70/HSP40 complex [93].

### 2.5. Analysis of the Interface Dynamics

Various tools are now available that have been tuned to process all-atom MD trajectories. These tools are developed to gauge changes in covalent (intra) and non-covalent (inter) interactions, as well as changes in the intrinsic dynamics of a biomolecular complex. These analysis tools usually come together with the simulation package used to create the relevant MD trajectory. In the case of GROMACS, for example, a vast collection of gmx scripts allow the user to calculate changes in interfacial backbone-related terms (such as dihedral angles), hydrogen bonds, salt bridges, specific interfacial distances, buried surface area, interfacial water molecules, and RMSDs/RMSFs [94]. The corresponding analysis package for NAMD is VMD (Visual Molecular Dynamics) [95], whereas for AMBER, it is CPPTRAJ/PYTRAJ (Amber Tools) [96]. Next to their own internal formats, all these tools can also work with any ensemble file, if recorded in the PDB format. Among these tools, gmx and CPPTRAJ are command-line-based, VMD is GUI-based, and PYTRAJ provides something in between, i.e., it runs on a Jupyter notebook with the NGL viewer molecule visualization option.

If the user requires an analysis that is beyond what is provided by these widely used simulation engines, then they can refer to cross-platform analysis tools, such as MDTraj and MDAnalysis [97]. Both tools are composed of various Python libraries to perform MD trajectory analysis. Besides their common analysis functions, they have tool-specific options, such as the calculation of fraction of native contacts (MDAnalysis) or an extended hydrogen bond analysis, where different h-bond definitions can be invoked (MDTraj). There are also approaches that utilize the libraries of MDAnalysis and MDTraj. One such tool, ProLIF, is specifically finetuned for calculating non-covalent interfacial contact types (hydrogen bond donor/acceptor, pi stacking, anionic, cationic, etc.) and their propensities throughout the whole trajectory [98]. A similar tool to ProLIF is Interfacea, another Python library, which provides interfacial non-bonded contacts, as well as OpenMM-based interaction energetics [99]. gRINN can also calculate residue-based interface energetics using a network-based approach [100]. The gRINN tool uses energy values calculated either by GROMACS or VMD to present residue-based energy changes and correlations through a GUI. ProDy and MD-TASK, two other network-based tools, allow us to calculate the essential dynamics of the whole complex, from which the interface region could be specifically investigated [101,102]. All the aforementioned tools for MD analysis are listed in Table 1.

As introduced above, to visualize MD analysis results, VMD has been widely used. In their very complete review on the visualization of biomolecular interactions [103], Agamennone et al. show how Visual Molecular Dynamics (VMD) [95] can be used to retrieve dynamic interface information from an MD trajectory of the spike RBD-ACE2 complex. More specifically, in this example, the stability of the contacts is measured by monitoring the occupancy of hydrogen bonds along time. Recently, a web-based tool, 3dRS, was published to provide non-experts with the opportunity to investigate MD trajectories and share their visual representations (https://mmb.irbbarcelona.org/3dRS/, accessed on 10 January 2022, ref. [104]). In their work, Bayarri et al. enable tracing of the trajectory details over a web browser by keeping the predefined structural annotations. Each visualization session has a unique link, which can be shared and reworked. This is achieved using NGLview and MDSrv tools that are built upon the MDAnalysis package [105,106]. The use of this tool holds great potential to provide a general understanding of the protein–protein interaction dynamics explored by MD trajectories.

## 3. *E Pur Si Muove!* How Can We Relate Protein Function to Interfaces Dynamics?

The dynamics of interfaces reflect fundamental properties of the protein–protein assembly such as the interaction specificity, the stability of the complex, or the kinetics of association and dissociation [107,108]. Notably, these properties can be directly linked to characteristics of the binding energy landscape, such as funneling, roughness or shallowness, and the dimensionality of the essential movements [109]. Although the roles of several key factors on association have been explored (e.g., [110,111,112,113,114,115]), a global view of how the characteristics of the interface dynamics influence specificity, association strength, or binding rates remains to be constructed [108]. Contrary to the generally accepted idea that strong binding affinity requires precisely defined interactions across the interface, a recent study associated a highly disordered interface and very strong binding affinity for the histone H1 bound to its nuclear chaperone prothymosin-α [116]. This suggests that extracting pertinent descriptors of the interface dynamics that would efficiently capture biologically meaningful assembly properties [117] is not straightforward. Yet, this knowledge is essential to understanding how association can be regulated via modifications of the pH, salt concentration, or small ligand binding [118], and more generally, to efficiently design PPI-targeting drugs. It is also important to evaluate the accuracy of protein complex modeling efforts.
Figure 2Protein interfaces take many shapes. Looking at their dynamic properties can bring us precious information on their function: (**a**) A protein complex with fully folded partners, target 29 from the CAPRI score-set [119], tRNA m7G methylation complex from yeast (pdb code 2vdu) with the catalytic unit Trm8 (in cyan) and its partner Trm82 (in magenta); (**b**) A protein complex comprising disordered regions, p150 unit from the eukaryotic initiation factor 4F (in magenta) folds upon binding the translational initiation factor A4 (in cyan), but its N-terminal tail remains disordered (pdb 1rf8); (**c**) A protein complex with active interfaces, cryo-EM structure of a microtubule fragment with GDP (in orange) bound at the interface between the tubulin α (in cyan) and β (in magenta) chains. (pdb 3j6f). All graphical representations were made with VMD [95].
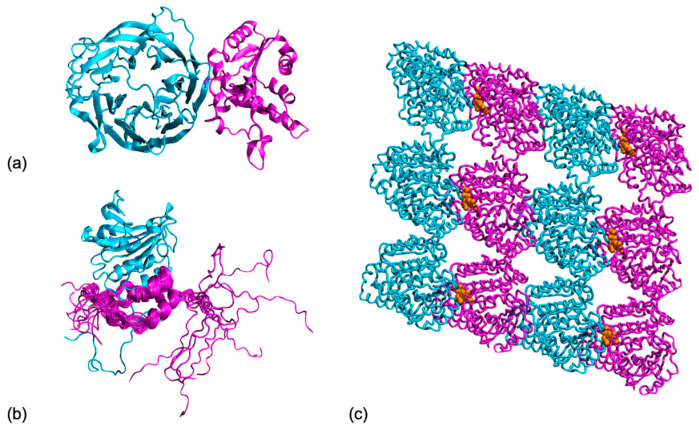


### 3.1. Interface Dynamics between Folded Partners

Addressing protein dynamics is a central problem in predicting protein–protein interactions, as the formation of a protein assembly is likely to induce conformational changes in the interacting partners. As a consequence, many docking algorithms will use conformational ensembles of the interacting partners to produce dynamically relevant structural models for the protein complex. Even so, these final complex models are still considered rigid objects during their quality assessment and ranking, and the dynamic properties of the generated interfaces are rarely investigated after docking, probably because of the high computational cost associated with a follow-up simulation at the atomic scale.

#### 3.1.1. Dynamics of the Isolated Partners

One key issue when attempting to model protein–protein interactions is the accurate location of the binding patches on the partner surfaces. To reveal the binding sites through the lens of protein dynamics, Kuttner and Engel used steered MD simulations to investigate protein backbone dynamics [120]. They show that some surface residues will form *stability patches*, i.e., residues presenting a reduced local mobility, and that their location overlaps with the protein–protein interface. Once the binding sites have been located on the protein surface, their dynamics can play a key role for biomolecular recognition. Fuchs et al. [31] performed MD simulations to investigate the specific binding of thrombin with various peptides, using the binding site dynamics to calculate the entropic contribution to binding affinity and predict specificity. The isolated proteins’ intrinsic dynamical properties are also tightly connected to their promiscuity, i.e., the ability to interact with different partners. Fornili et al. [121] performed a large-scale study on 250 proteins extracted from the PDB. They found a significant difference between the conformational flexibility of monopartner and multipartner residues, the latter being, on average, more flexible.

In a “pre-docking” perspective, the protein dynamics can also be accounted for using alternative conformations of the partners. The PRISM (Protein Interactions by Structural Matching) tool [30,34,122] uses conformational ensembles for each partner to predict complex structures and build protein–protein interaction networks. NMA has also proved to be a useful tool for sampling the conformational variability of the interacting partners. In the SwarmDock algorithm [123,124], the multiple backbone conformational states of the protein partners are generated using low-frequency modes, resulting from NMA on an elastic network model.

#### 3.1.2. Interface Dynamics within the Protein Complex

When analyzing the dynamics of interfaces, a first source of information can be obtained via docking simulations since these methods extensively sample possible binding geometries. Even rigid-body sampling can provide a useful overview of the interface properties. In its selection process, the ClusPro server explicitly includes the relative sizes of the clusters of lowest-energy predicted structures, identified during Fast Fourier Transform sampling [125]. Inclusion of this entropy-related criterion is one of the strengths of the method, which contributes to its success in the last Capri rounds [126]. Further, in a study that used both rigid-body FFT sampling and the RosettaDock Monte Carlo minimization algorithm, Kozakov and collaborators could characterize the energy landscape of encounter complexes in the vicinity of more than 40 complexes. In this way, they showed that association occurred along a small number of preferred pathways [109]. Readers wishing to analyze docking-generated conformational ensembles can consult Table 2, which includes the most commonly used protein–protein docking servers (refs. [125,127,128,129,130,131,132,133,134,135,136].

MD simulations performed after docking are mostly used for the refinement of the interfaces. In this case, though, the algorithms use very short time scales to save time. For example, the HADDOCK [58,137] procedure includes a final refinement step of MD simulations in explicit solvent or in vacuum. However, in this case, the standard duration of the heating, sampling, and cooling phases lasts less than 10 ps, providing insufficient time to investigate the dynamic properties of a protein–protein interface. Even if the interacting partners are considered to be *rigid*, i.e., when the conformational changes induced by the binding process lead to a root mean square deviation (RMSD) that is below 1 Å, the protein interface remains a dynamic object, and trajectories from short (50 ns) MD simulations can be used to study the effect of the solvent on the complex surface. For example, Visscher et al. used this approach to show how the hydration layer contributes to the complex stability [32].

Recently, research groups began using all-atom MD simulations to investigate the stability of structural models produced by docking to assess their quality. Radom et al. [138] used short-timescale (below 100 ns) MD trajectories, starting from near-native models of complexes produced by RosettaDock [85]. They showed an increased stability of the correctly docked complex compared to incorrectly docked complexes, as the latter tend to unbind upon increasing the temperature in the simulation box. Jandova et al. [139] used a combination of MD simulation and machine learning tools to try and distinguish native from non-native docking models. Again, native models show higher stability in almost all measured properties, including the criteria traditionally used for scoring complex models in CAPRI, namely the ligand and interface RMSDs, and the fraction of native contacts from the reference experimental structure of the protein complex. Prévost and Sacquin-Mora also questioned the relevance of the current CAPRI criteria, as they are based on a single static reference structure [140]. In their work, they ran MD simulations both on the experimental reference structure and various near-native models for three CAPRI targets (see, for example, T29 in Figure 2a) and showed how using dynamic criteria (based on the trajectory analysis and not on a single structure) can impact the models’ ranking, as they display different stabilities over time.

MD simulation of protein complexes can also help identify the residues playing a part in partner selection in the signaling pathways. To this end, Van Wijk et al. [35] investigated the dynamics of E2–E3 interactions, using both wild-type and mutant structures. They showed how a dynamic salt-bridge network controls the interaction selectivity via the modulation of side-chain conformations. As a more recent example, Nicoludis et al. explored how protocadherins specifically find their partners to polymerize, an essential step during neuronal development. For this, they investigated interface dynamics through MD simulations and combined this information with evolutionary couplings [37]. Finally, Karakulak et al. demonstrated that comparative modeling and simulation of three paralog complexes, taking a central role in cellular signaling (the TAM receptor pathway), can deliver the partner selecting residues of the receptors [141]. Like in the E2–E3 case, salt bridges were shown guide the partner selection.

Interestingly, numerous studies on protein complexes that use MD simulation do not focus on the interface dynamics per se, but on other properties. For example, MD can be used to determine the binding free energy in the complex. Hou et al. [142] performed CG-MD simulations with the MARTINI protein force-field on near-native models from the CAPRI Score_set [119] to evaluate their binding free-energy. These free-energy values were then used as a scoring method to rank the docked structures. The GroScore approach [143], developed by Perthold and Oostenbrink, relies on MD simulations to produce short (5 ns) unbinding trajectories that are used to compute binding free energies. These free energies were then used for the scoring of docked protein poses extracted from the CAPRI Score_set benchmark [119].

Finally, NMA has been coupled with several docking approaches to model the conformational changes that occur within the partners upon binding. Schindler et al. developed iATTRACT [144], where the interface residues will move following a NMA-generated harmonic potential, in order to refine models produced by an initial rigid-body docking step. In iNMA [145], Frezza and Lavery used the protein partners internal coordinates (namely the torsion angles) to capture large conformational changes in the partners and generate structures closer to the partner’s bound state when starting from their unbound shape. One should also note that changes in the interface dynamics can impact distant residues that are located away from the binding site. Eren et al. used a combination of ENM and NMA on the KRas4B GTPase [146] to reveal how the protein dynamics depend on its interaction partner. The dependency of the interface dynamics on the protein partners was also observed by Paul et al. in their study, where they used all-atom MD simulations on ubiquitin bound to two different proteins [147]. Although the two complexes share the same binding site on the ubiquitin side, both the dynamics of the interface residues and the global ubiquitin dynamics are impacted differently depending on the interaction partner.

### 3.2. Interfaces Dynamics within Disordered Partners

Intrinsically disordered proteins (IDPs) or regions (IDRs) can be found in nearly a third of the human proteome [148], and IDPs form around 10% of the 10,000 structures that are deposited annually in the PDB [149]. Disorder is an important feature of protein interactions, as disorder-to-order transitions are estimated to be present in 42–75% of binding events [29]. Over recent decades, the key role played by IDPs and IDRs in numerous cellular processes has become increasingly clear [33,150,151]. This protein group is estimated to harbor around 25% of disease associated missense mutations [152], thus making IDPs/IDRs a central therapeutic target. As protein assemblies comprising IDPs or IDRs are very likely to form *fuzzy complexes*, where one or both partners in the interaction will retain some disorder [153,154], investigating the dynamics of such complexes appears to be an unavoidable step for understanding their function [50,155]. IDPs can also undergo folding-upon-binding transitions when assembling with another protein partner [156] (see Figure 2b). In that case, the initial conformational disorder in the unbound partner has been shown to have a complex impact on the binding kinetics, as it can either increase or decrease the association/dissociation rates between the partners [157,158]. Again, keeping a dynamic view of the protein interface is essential to properly describe and understand these assemblies.

### 3.3. Active Interfaces within Molecular Machines

So far, we have addressed the dynamic properties of unique, well-defined interfaces. We have discussed how the dynamics of these interfaces and the conformational space they sample at room temperature can be characterized. We now turn to the cases where interface dynamics lead to transitions to alternative assembly formation. Indeed, many proteins present ubiquitous binding modes [18], and these proteins can modify their binding geometries in response to perturbations that include changes in the chemical environment (salt concentration, pH, …), mechanical forces, or chemical reactions. Interface switching can have a functional role, notably, in macromolecular motors. It can constitute an essential cog of the transformation process from chemical energy to mechanical energy.

Macromolecular motors are protein complexes with mechano-chemical properties; they can, for example, act as helicase, translocase, recombinase, nuclease, or protease to process other macromolecules, either DNA or proteins [159]. These systems transform chemical energy, often the energy released by the hydrolysis of NTP cofactors, into translational or rotational movements. To this aim, they undergo cooperative conformational changes, which involve interface reorganizations. In other words, the binding equilibrium between motor components is displaced by a chemical reaction towards interface modification or even towards dissociation.

Within this context, we discuss here the case of oligomeric molecular motors. These complexes present several instances of the same monomer–monomer interface. As a consequence, even small changes in the interface can have a huge impact on the overall shape of the oligomer [160]. In the absence of external perturbation, oligomeric assemblies do not substantially modify their global shape in spite of the binding interfaces constantly exploring the available conformational space as a result of thermal motions. Molecular dynamics simulations on the nucleoprotein filament of RecA recombinases have shown that during this exploration, interfaces typically conserve 70% to 80% on average of their residue contacts [161]. However, small deformations in individual interfaces can produce substantial modifications of the overall assembly. Microtubules provide another good example of this interface variation amplification effect. They are tubular-shaped assemblies of α,β-tubulin dimers that constitute an important part of the cytoskeleton. Microtubules hydrolyze GTP cofactors bound at the longitudinal interface between consecutive dimers (Figure 2c), which only marginally modifies the geometry of these interfaces [162]. Small reorganization of the interface residues results in a slight compaction of the dimer interface that displaces the binding equilibrium in a way that favors a slight curvature. When amplified by its repetition over a whole protomer (longitudinal assembly of α,β-tubulin dimers), this slight curvature destabilizes the protomer assemblies and finally results in a catastrophic disassembly of the microtubule. Cycles of assembly/disassembly of microtubules generate a force that is used in mitosis to displace the duplicated chromosomes.

In cyclic ring motors, where a limited number of monomers are assembled, consequences of interface changes are less spectacular. Nonetheless, they are of particular interest, as ring closure requirements necessitate that interface changes are non-uniformly distributed along the ring. The simultaneous presence of different binding geometries that break the ring motor symmetry has first been detected by electron microscopy [163]. In such complexes, the interfaces can be modified in a coordinated manner, which is coupled with the progression of the hydrolysis cycle along the ring: at a given time, the particular state of each interface is defined by the state of the ATP hydrolysis process. State progression along the ring has been characterized by single-molecule experiments in a series of systems (see [164] and references herein), revealing different ways of coupling between the hydrolysis states, which correspond to different kinetic regimes of the motor activity. These observations needed an interpretation in terms of interface dynamics, which has been offered by a molecular dynamics study of Ma and collaborators [165]. They simulated the transition between different states of ATP hydrolysis within the ring-shaped ATPase motor Rho, which translocates an RNA strand across membranes. This work notably showed how the changing network of interactions, within protein interfaces and between proteins and the processed RNA strand, coordinates the strand translocation, the positioning of residues involved in ATP hydrolysis, and the accessibility of ADP for its replacement by ATP. The study enabled characterizing the free energy of transition between the different substates involved in the motor mechanism.

In the absence of a strong coupling such as the one found in Rho, ATP hydrolysis may induce breaks in the ring. Indeed, such breaks have been observed that resulted in the transformation from the ring assembly into a spiral assembly. Examples of oligomers that can assemble either as rings or spirals have been highlighted in cases such as the DnaB hexamer, which acts during replication [163]. Both forms may be involved in the oligomer function. Another impressive visualization of the dynamic transition between ring-shaped and spiral assemblies of ClpB, a protein hexamer disaggregation machine, was captured via high speed atomic microscopy [166]. It was observed that the frequency of such transition events is directly linked to the concentration of ATP in the sample.

Finally, reported changes in binding geometry in response to perturbations in oligomeric systems show a large range of variations. We have seen that in microtubules, the change expands on minor rearrangements [162]. In cyclic systems such as the Rho hexamer [165], the interface where ATP hydrolysis takes place loses up to 60% of the residue-residue contact interactions with respect to its neighboring interfaces, but the remaining interfaces conserve at least 70% and up to 90% of the contact interactions (analysis performed with the PTools/Heligeom tool [160], not shown). In the open, spiral filament of RecA proteins active in homologous recombination, the totality of interface residue–residue contacts between rigid regions are lost when ATP is hydrolyzed [160] (Figure 3). In this instance, the transition from one interface to the other involves a 20 Å relative displacement of the two monomers that obviously will require more than thermal motion assisted by the local interface rearrangement due to the Pi dissociation. Concerted movements, probably coordinated by the DNA strands bound to the filament interior, are expected to play a key role in this intriguingly large interface rearrangement.

## 4. Conclusions and Perspectives

The dynasome, i.e., the ensemble of mobility patterns that can be presented by a protein, has been considered a central element in our understanding of proteins for over a decade, as it was often described as the missing link between structure and function [167]. This led to the creation of the MoDel (for Molecular Dynamics Extended Library, http://mmb.irbbarcelona.org/MoDEL/, accessed on 30 November 2021) dataset, which comprises over 1700 trajectories, from atomistic MD simulations of soluble monomeric proteins [168]. Numerous additional databases storing data from biomolecular simulations now exist. They deal with specific protein families such as GPCR (GPCRmd, https://submission.gpcrmd.org/home/, accessed on 30 November 2021) [169], or specific systems, such as the BioExcel-CV19 platform (https://bioexcel-cv19.bsc.es/#/, accessed on 30 November 2021), which provides web access to atomistic-MD trajectories for macromolecules involved in the COVID-19 disease. However, we are still missing similar initiatives for protein–protein assemblies. This is of importance, as such initiatives play a key role for the development of artificial intelligence and machine learning (AI/ML)-based approaches, since they rely on the availability of large datasets. The year 2021 has shown very promising work using AI/ML strategies for the prediction of protein assemblies [170], notably when investigating SARS-CoV-2-related proteins [171,172], thus highlighting the need for similar large repositories dedicated to protein interfaces.

As high-performance computing facilities now render long simulations increasingly accessible worldwide, it is high time for us to address the fourth dimension of macromolecular assemblies, namely time, and start investigating their dynamic properties in a more systematic fashion, if we want to decipher the protein *social network* in the cell.

## Figures and Tables

**Figure 1 molecules-27-01841-f001:**
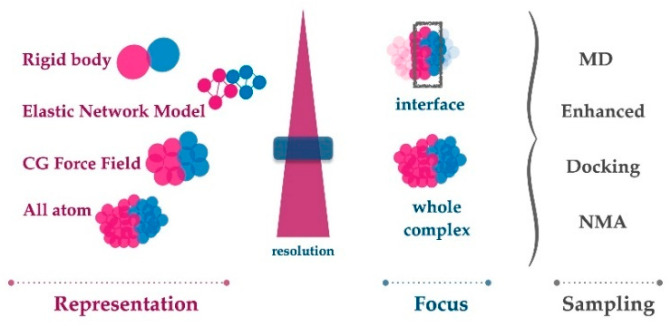
The accurate modeling of protein interface dynamics relies on three choices: system representation, system focus, and the sampling algorithm. If the level of information is sought at the atomic scale, then all atom representation should be selected. Depending on the resources that can be invested, the dynamics of the whole complex or only the interface could be chosen as the focus. In such a situation, classical MD would generate the finest level of sampling. Though, for bigger systems or shorter computing times, faster enhanced sampling methods could be used. If larger-scale motions are expected, then coarse grain (CG) force fields or elastic network models could be used as system representations. Those can be sampled with any of the sampling methods listed. Finally, in case the binding mechanism is investigated, rigid body minimization driven docking could deliver several solutions that could represent encounter complex formation. This solution set could also tell us how different solutions in a well-defined interface can fluctuate, thus indirectly reporting on the interface dynamics.

**Figure 3 molecules-27-01841-f003:**
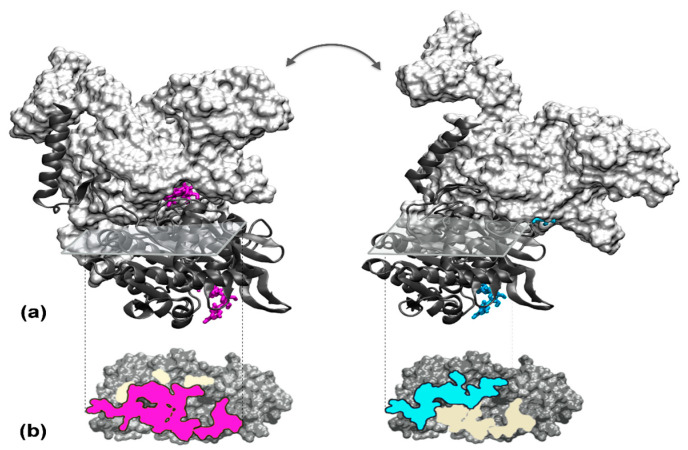
Cofactor-dependent RecA-RecA interface changes. (**a**) Two interacting RecA monomers are represented in the presence of ATP (left) or ADP (right). In both views, the top monomer is in surface mode, colored white, and the bottom monomer is in cartoon mode, colored dark grey. The ATP and ADP molecules are represented in licorice, respectively, in magenta and cyan. The bottom monomers are represented in the same orientation. (**b**) Projection of the ATP (left, interface in magenta) and ADP interface (right, interface in cyan) on the surface of the bottom monomer. In both views, the regions that are present in the other interface are in light yellow. Although the two interfaces overlap, amino acids that are present in both interfaces interact with different amino acids in the opposite partner.

**Table 1 molecules-27-01841-t001:** List of the described MD analysis tools that can be used to dissect interface dynamics.

Tool Name	Related Link (All Sites Were Accessed on 28 January 2022)
GROMACS	https://manual.gromacs.org/documentation/2021/reference-manual/analysis.html
VMD	https://www.ks.uiuc.edu/Research/vmd/
PYTRAJ/CPPTRAJ	https://amber-md.github.io/pytraj/latest/index.html
MDTraj	https://www.mdtraj.org/1.9.5/index.html
MDAnalysis	https://www.mdanalysis.org
ProLIF	https://github.com/chemosim-lab/ProLIF
interfacea	https://github.com/JoaoRodrigues/interfacea/tree/master
gRINN	grinn.readthedocs.io
ProDy	http://prody.csb.pitt.edu/
MD-TASK	https://md-task.readthedocs.io/

**Table 2 molecules-27-01841-t002:** List of automatic docking web servers.

Server Name	Web Site (Accessed on 28 January 2022)	Conformational Ensemble Retrieval	Reference
ClusPro	https://cluspro.org/	10 most populated low energy clusters, irmsd > 9 Å	[125]
PatchDock	http://bioinfo3d.cs.tau.ac.il/PatchDock/	Up to 100 top ranking candidates; clustering cutoff adjustable	[127]
GRAMM-X	http://vakser.compbio.ku.edu/resources/gramm/grammx/	Up to 300 lowest energy conformations	[128]
RosettaDock	http://rosettadock.graylab.jhu.edu	1000 decoys can be downloaded	[129]
MDockPP	https://zougrouptoolkit.missouri.edu/MDockPP/	Up to 3000 generated geometries; clustering cutoff adjustable	[130]
HADDOCK	https://wenmr.science.uu.nl/haddock2.4/	All generated geometries can be downloaded	[131]
pyDockWEB	http://life.bsc.es/servlet/pydock	Top 100 lowest energy conformations	[132]
ZDOCK	https://zdock.umassmed.edu/	Top 10 lowest energy conformations; possibility to retrieve top 500	[133]
InterPred	http://bioinfo.ifm.liu.se/inter/interpred/	No conformational search (template-based)	[134]
HDOCK	http://hdock.phys.hust.edu.cn/	Top 100 lowest energy clusters, lmrsd > 5 Å	[135]
LZerD	https://lzerd.kiharalab.org/	Up to 50,000 generated geometries	[136]

## Data Availability

Not applicable.

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
