# Peer review of "Modeling the Dynamics of Protein–Protein Interfaces, How and Why?"

_molecules, 2022, doi:10.3390/molecules27061841_

Round 1

Reviewer 1 Report

The work is devoted to a topic (assessment of the dynamics of protein-protein interfaces using different computational and experimental approaches) is highly demanded by a large category of scientists and contains a significant number of interesting references.

However, since this is not the result of a specific experimental or theoretical work of the authors, but rather an analytical review, one would like to see not only a listing of who did what, but an analytical analysis of the described methods and examples. And since the review is mostly devoted to methods, its goal is to help researchers in choosing a method for solving a specific problem related to the structural study of protein-protein complexes.

Therefore, I believe that the authors could significantly improve their work by strengthening the analytical part and providing the necessary basic information about the diversity of protein complexes, as well as experimental and computational methods for their study.

Namely,

1) Give a more detailed and clear description of different types of complexes: from simple complexes to multicomponent molecular machines. For simple ones, classify by affinity and stability (lifetime) from high affinity antigen-antibody through receptor-ligand to short-lived signal-transduction and enzyme-substrate complexes.

(this info is in the introduction and somewhere in the results, but not very clear and complete).

2) Show correlation between the types of complexes with the appropriate experimental methods of structural biology (X-ray crystallography + SAXS, NMR, CryoEM) and their advantages and disadvantages, indicating what and when is applicable at least in the general case. Ideally, to represent it schematically

3) In chapter 2.4, one would like to see a discussion about the contribution of experimental biophysical techniques that are not methods of structural biology in the study of interfaces.

I mean, namely mass spectrometry in combination with protein-crosslinking and limited proteolysis, but also calorimetric and spectral methods et cetera.

It seems that the whole chapter 2.4 needs to be revised, because there are too many uninformative (and generally unjustified and incomprehensible generalizations), for example

The structure and dynamics of interface residues, or the lifetime of local contacts, can be obtained via X-ray crystallography, NMR, CryoEM or FRET [28]. - Perhaps in [28] this connection was explained for a particular case, but apriori listing the main methods of structural biology + FRET looks strange. (This is not the only example)

4) In chapter 2.5, Table 1, in addition to the name of the program/software package and the Internet address, add references on the original source or latest update of the programs, as well as the references from the text describing the usage of the program and, if possible, a brief description of the subject of the study (if there is one) or an indication that this is a review ... If it is difficult to fit in one table, then distribute the information between 2 tables: one in the text, another in a supplement.

If possible, add to the text arguments about the advantages and disadvantages of using certain programs to study different types of complexes. Comparisons of similar programs among themselves (for example, Gromax or Amber - what to choose?) would also be interesting for readers who are just starting to work in the field of structural biology.

5) Chapter 3 clearly asks for some kind of analytical table on docking methods and / or computational approaches described in the text.

6) The meaning of figure 1 is not clear. Neither It is not clear how the figure relates to its general caption (1), nor the connection of the presented pictures with the text in the article related to the figure 1 (2).

(1) Why these pictures were chosen? And how do they illustrate the variety of interfaces?

(2) How does Figure 1a illustrate the text below?

“In their work, they ran MD simulations both on the experimental reference structure and various near native models for three CAPRI targets (see Figure 1a), and showed how using dynamic criteria (based on the trajectory analysis and not on a single structure) can impact the models ranking, as they display different stabilities along time.” (The note applies to other panels as well).

7) The conclusion has nothing to do with the main text.

8) Graphic abstract – it is not very clear, it is just a picture or a movie / animation?

Then where can one watch it?

If this is a static image, it does not reflect the caption text: Monitoring the contacts between partners in the protein-protein interface formed by colicin E9 endonuclease and its bacterial inhibitor (pdb 2wpt) along time can help distinguish near-native docking models.

Minor

The authors note an interesting pattern, but do not further discuss this very interesting topic: “Interestingly, numerous studies on protein complexes that use MD simulation do not focus on the interface dynamics per se, but on other properties”.

Reviewer 2 Report

This is an excellent and well written review on computational approaches to study the dynamics of protein-protein interactions. The review is also timely because deep learning models (such as AlphaFold2) significantly improved our structure modeling capabilities, while the dynamics and consequently the function remain behind.

I have one suggestion: it would be excellent to accompany section 2 with a schematic figure that can guide the reader through the advantages/disadvantages of the available methods.

Author Response

Authors response: Figure 1 has been added at the beginning of section 2 to summarize the
different possible strategies when modeling protein interface dynamics

Reviewer 3 Report

This is an excellent review of the state-of-the-art computational methods to study protein-protein interactions. Both all-atom and coarse-grained methodologies for calculation, as well as methods of analysis of the results, are discussed.  The methodologies are illustrated with examples. The review is very clearly written and will be useful both to beginners and advanced researchers in the field. Optionally, the Authors might want to write a short section of the servers available for protein-protein docking, along with the respective web page addresses, e.g., the HADDOCK server (all-atom approach, https://wenmr.science.uu.nl/) the CABS-dock server (CG approach, http://biocomp.chem.uw.edu.pl/tools/cabs-dock), UNRES server (CG approach, https://unres-server.chem.ug.edu.pl/login/?next=/). The papers that describe some of the server, e.g., HADDOCK, are in the reference list, but listing them explicitly in the paper would be useful for those who are looking for web-based protein-protein docking tools.  

Author Response

Authors response: This is a very good point. Following the suggestions of Refs. 1 and 2 a
table listing automatic docking webservers has been added to section 3.1.2.

Round 2

Reviewer 1 Report

In general, I am satisfied with the changes made to the text and the answers provided. However, I want to note that the new picture and table, as well as one of the old tables, are inserted into the text incorrectly. At least that's how it is displayed for me.

In addition, I strongly recommend  to think about a more interesting graphic abstract that reflects the essence of the work.

Author Response

We checked the positions of all figures and tables in the manuscript and they should now be correctly positioned. Note that the appearance of the .docx file can vary depending on the software used to open it, but the pdf file should always be correct.